# Practical Modelling of Mixed-Tailed Data with Normalizing Flows

**Saba Amiri**                                                                                     *s.amiri@uva.nl*
*Informatics Institute*
*University of Amsterdam*

**Eric Nalisnick**                                                                                 *nalisnick@jhu.edu*
*Department of Computer Science*
*Johns Hopkins University*

**Adam Belloum**                                                                                   *a.s.z.belloum@uva.nl*
*Informatics Institute*
*University of Amsterdam*

**Sander Klous**                                                                                   *s.klous@uva.nl*
*Informatics Institute*
*University of Amsterdam*

**Leon Gommans**                                                                                   *leon.gommans@klm.com*
*Air France–KLM*

**Reviewed on OpenReview:** *https: // openreview. net/ forum? id= uphsKDj0Uu*

## Abstract

Capturing the correct tail behavior is difficult, yet essential for a faithful generative model. In this work, we provide an improved framework for training flows-based models with robust capabilities to capture the tail behavior of mixed-tail data. We propose a combination of a tail-flexible base distribution and a robust training algorithm to enable the flow to model heterogeneous tail behavior in the target distribution. We support our claim with extensive experiments on synthetic and real world data.

## 1 INTRODUCTION

Real-world data often show mixed tails - both heavy (representing 'black swan' events, seen in communication networks traffic and actuarial risk (Wang et al., 2006; Afify et al., 2020)) and light (indicating events within narrow outcome ranges, seen in certain pricing models and extreme engineering events (Singh & Gor; Jamissen et al., 2022)). Capturing tail behavior accurately is vital in data synthesis, especially in sectors such as healthcare, where mis-estimation can cause significant inaccuracies in analyses based on the synthesized data, leading to misinferred risks (Ibragimov et al., 2015).

Normalizing flows (Papamakarios et al., 2021) present an attractive model for mixed-tailed data synthesis. This is due to the fact that the tail behavior of the base distribution directly and transparently affects the tail of the generated data (Jaini et al., 2020). Other classes of deep generative models do not provide such an explicit mechanism to control their tails. Figure 1 demonstrates the impact of mis-specification of parameterization of base density on the capabilities of normalizing flows on capturing the tail behavior of targets with different tail behaviors.

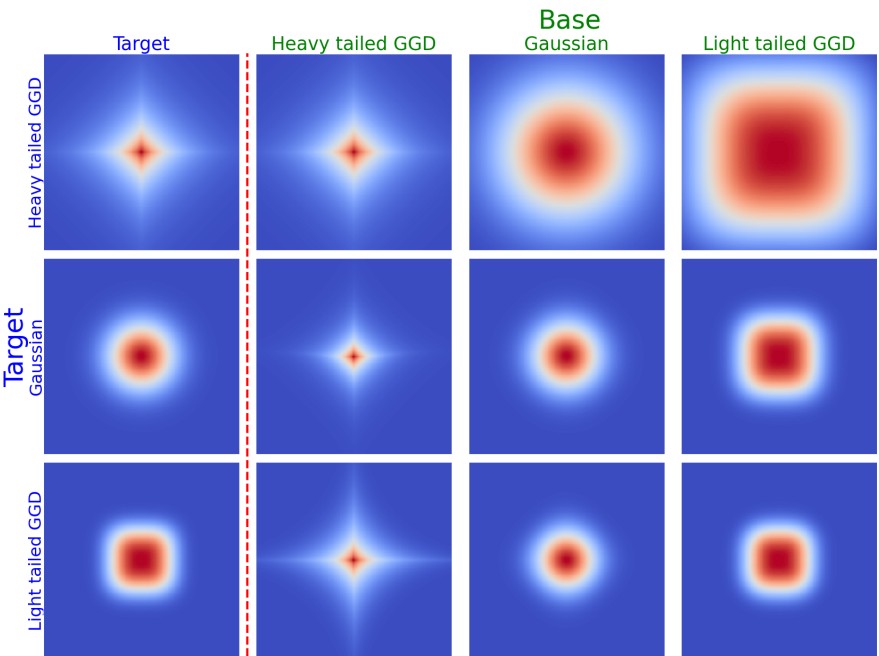

Figure 1: The impact of mis-specification of base density on the capabilities of normalizing flows to capture the tail behavior of the target density. We train an RNVP on three short, normal and heavy-tailed Generalized Gaussian (GGD) targets (columns 2-4) with short, normal and heavy-tailed GGD base densities (rows 1-3). We plot the probability density of the plot. Mis-specified parameterization of base leads to flow being only able to estimate the target with the same tail behavior as its base density accurately

The task of synthesizing mixed-tail data using normalizing flows is an intriguing research domain within generative modeling. Previous methods (e.g. (Jaini et al., 2020; Laszkiewicz et al., 2022)) select the base distribution to match the target tails. One potential concern with applying these methods to mixed-tail targets is that the base distribution might be overly conditioned on the estimated or perceived tail of the target. This makes the flow model brittle, as will be discussed in Section 3. We also posit that mixed-tail data can negatively impact model gradients during training regardless of the choice the family and tail behavior of the base density.

In this work, we focus on improving the flow model for mixed-tail targets. In contrast with previous methods which select the base density, we make the base density of the flow adapt to the target tail characteristics across a broad family by making it flexible enough to capture the tailedness of the target through a flexible mixture of *Generalized Gaussians* without binding the base to the tail properties of the target distribution. To make the flow model itself capable of capturing the tail of the target distribution we make the training process robust to heavy-tailed gradients. By making the Monte Carlo estimator robust we stabilize the training process and enable model to capture the tail behavior of the target distribution and admit gradient-based adaptation of the tail behavior. Following prior works in this area (Jaini et al. (2020); Laszkiewicz et al. (2022)), we validate our method on a range of data synthesis tasks, benchmarking it against other state of the art tail-adaptive flows and empirically show the advantage of our method for mixed-tail targets. We show general improvements in capturing the tail behavior of mixed-tail targets while maintaining high general utility.

Our contributions include:

- Highlighting the problem with pre-hoc *selection* and *estimation* of base density tail and proposing a flexible tail-adaptive mixture for base density to fix these issues.

- Discussing the problem with maximum likelihood training of normalizing flows on mixed-tailed targets and propose a robust method to counter the impact of heavy-tailed gradients.

- Empirical results showing generally favorable performance compared to other tail-adaptive flow methods.

## 2 BACKGROUND

**Notation** We denote univariate and multivariate random variables with bold letters and observations with non-bold letters. We denote data by $\mathbf{x}$ and its observation by $x$. The $k$th component of $\mathbf{x}$ is denoted by $x_k$.

### 2.1 Characterizing Tail Behavior

A random variable $X$ is said to follow a *heavy-tailed* distribution if its tail is not exponentially bounded. This can be formally defined using the moment generating function (MGF): $M_X(t) = E[e^{tX}]$. A distribution is heavy-tailed if there exists some $t > 0$ such that $M_X(t)$ is infinite or does not exist. An example of this category is the Pareto distribution with scale parameter $\gamma > 0$, for which the MGF is $M_X(t) = \frac{\gamma}{\gamma - t}$, which is undefined for $t \geq \gamma$.

The tails of a *light-tailed* random variable $X$ decay sub-exponentially. The Gumbel distribution, with location parameter $\mu$ and scale parameter $\beta > 0$, is an example of a light-tailed distribution as its survival function $S_X(x) = e^{-e^{-(x-\mu)/\beta}}$ indicates a sub-exponential decay for large $x$.

In many real-world scenarios, we encounter data with mixed tail properties. We define such data as having *mixed tails* if some of its marginal distributions exhibit heavy tails and others light tails. This underscores the multifaceted nature of real-world data, where different variables or features may be governed by distinct distributional properties.

**Tail Index Estimation** The tail behavior of a probability distribution is primarily characterized by their *tail index*. The tail index, which we denote as $\alpha$, is defined as the exponent in the power-law tail of the distribution:

$$F(x) = P(X \leq x) \sim 1 - x^{-\alpha}$$

where $P(X \geq x)$ is the probability of the random variable $X$ being greater than or equal to a certain value $x$, and $F(x)$ is the cumulative distribution function of $X$. As $\alpha$ approaches zero, the distribution follows a power law to a greater degree. *Hill's estimator* (Hill, 1975) is a widely-used method for estimating the tail index. It is defined as follows:

$$\hat{\alpha}_H = \left[ \frac{1}{k} \sum_k^{i=1} (\log(X_{n-i+1}) - \log(X_{n-k})) \right]^{-1}$$

Here, $n$ represents the sample size, $k$ is the number of upper order statistics, and $X_{n-i+1}$ denotes the $i$th largest observation in the sample. The parameter $k$ is of crucial importance: a large $k$ inflates the variance of the estimator, while a small $k$ increases the bias (Fedotenkov, 2020).

### 2.2 Normalizing Flows

*Normalizing flows* (NFs) (Tabak & Turner, 2013; Rezende & Mohamed, 2015; Papamakarios et al., 2021) are models represented by a sequence of invertible transformations that warp a simple base distribution (such as a standard normal) into a richer target distribution. The transformations are chosen such that their (log) Jacobian determinant is easy to compute, allowing for efficient computation of the likelihood. Let $\boldsymbol{u} \sim p_u(\boldsymbol{u})$ denote the base density, and let the sequence of invertible transformations be denoted $T_\phi = T_{N-1} \circ \ldots \circ T_0$, where $\phi$ denotes the neural network parameters. We can evaluate the density of observation $\mathbf{x}$ by applying the change of variables formula:

$$p_\phi(\mathbf{x}) = p_u \left( T_\phi^{-1}(\mathbf{x}) \right) \left| \det J_{T_\phi^{-1}}(\mathbf{x}) \right|$$

where $J_{T_\phi^{-1}}$ is the Jacobian matrix of the inverse transformation. Due to the availability of the exact density function, maximum likelihood training can be carried out as usual:

$$\mathcal{L}(\boldsymbol{\theta}) = \mathbb{KL}\left[ \, p^*(\mathbf{x}) \parallel p_{\mathrm{x}}(\mathbf{x};\boldsymbol{\theta}) \, \right] = -\mathbb{E}_{p^*}\left[ \log p_{\mathrm{u}}\left( T^{-1}(\mathbf{x};\boldsymbol{\phi}); \boldsymbol{\psi} \right) + \log|\det J_{T^{-1}}(\mathbf{x};\boldsymbol{\phi})| \right] + C \qquad (1)$$

where $C$ is a constant, $p^*(\mathbf{x})$ denotes the true distribution of the data and $\boldsymbol{\psi}$ and $\boldsymbol{\theta}$ represent the parameters of the base distribution $p_u$ and the collective parameters of the model respectively. In practice, the expectation is calculated via a Monte Carlo approximation using the training set.

## 3 CHALLENGES OF TRAINING TAIL-ADAPTIVE NORMALIZING FLOWS FOR MIXED-TAIL TARGETS

Reviewing literature indicates that to adapt a flow model's base density to the target tail, existing methods either incorporate assumptions about tail behavior into the model through a specific base density selection, as in Jaini et al. (2020), or directly estimate the target's tail behavior, as in the marginal adaptive base method by Laszkiewicz et al. (2022). Furthermore, previous research (Behrmann et al., 2021) shows flow-based models suffering from the exploding/vanishing gradients problem, which will be exacerbated in presence of tailed mini-batches during training. We explore the limitations of these approaches in modeling mixed-tailed data.

### 3.1 Challenges in Adjusting the Tail Behavior of the Flow Model's Base Density

**Pre-Hoc *Selection* of Base Density Tail Behavior for Mixed-Tail Targets**  Real-world data may have tail behavior that deviate from any specific choice of base distribution family such as having a multi-modal mixed-tail distribution. Therefore, choosing a specific heavy-tailed parametric distribution family as the flow base as suggested by Jaini et al. (2020) a priori will inherently restrict the range of possible tail shapes and asymptotic decay rates that can be represented. The other issue with this method for mixed-tail targets is based on the fact that normalizing flow transforms on top of the base distribution can either alter its tail behavior significantly or not converge due to exploding/vanishing gradients. Thus to make the flow robust, the tail modeling capabilities must come more from the flows rather than the base distribution alone.

**Pre-Hoc *Estimation* of Base Tail Index for Mixed-Tail Targets**  Adapting the tail behavior of the flow's base density directly on the target distribution is another way to make the base distribution tail adaptive, with a prominent example being Marginal Tail Adaptive Flows (mTAF) by (Laszkiewicz et al., 2022). mTAFs calculate the tail index of each marginal of $x$. Since tail estimation is notoriously difficult, Laszkiewicz et al. combine the results from three different tail index estimators. The base density for light-tail marginals is set to a Gaussian and a univariate Student's t with a degree of freedom equal to the estimated tail index for heavy tail marginals.

The limitation of these methods for modelling complicated mixed-tail targets is their reliance on estimated tail index of the target distribution. Tail index estimators generally suffer from high variance and sensitivity to tuning parameters, especially for small sample sizes or multidimensional settings. For example, a major drawback of Hill's estimator is its sensitivity to the choice of $k$. Mason (Mason, 1982) demonstrated that $\hat{\alpha}_H$ is inconsistent if $k$ stays fixed as $n$ approaches infinity.

### 3.2 Challenges in Maximum Likelihood Training of Flow Models for Mixed-Tailed Targets

Regardless of the choice for the base distribution and its tail properties, we argue that it is simply not enough to enable the flow to be trained properly and capture the tail behavior of a mixed-tail target distribution. The reason is unstable training due to increased variance of the gradients when the target is heavy-tailed.

Consider the likelihood term in the maximum likelihood training objective (Equation (1)). Let $\boldsymbol{u} = T^{-1}(\mathbf{x};\boldsymbol{\phi})$ be the inverse transformation of the observation $\mathbf{x}$. The term $\log p_{\mathrm{u}}(\boldsymbol{u})$ in the likelihood involves the logarithm of the base density evaluated at $\boldsymbol{u}$. In the presence of heavy tails, the density can have slower-than-exponential decay, resulting in higher likelihood contributions from extreme values of $\boldsymbol{u}$ (and consequently, $\mathbf{x}$). Similarly,

$\log |\det J_{T^{-1}}(\boldsymbol{u}; \boldsymbol{\phi})|$ can become large in magnitude when $\boldsymbol{u}$ falls in the tails of the distribution since the Jacobian determinant accounts for the local expansion or contraction of the transformation and heavy tails can amplify this effect, leading to large values of the determinant.

## 4 METHOD

Our approach to tail adaptiveness in any normalizing flow hinges on: a) ensuring the base distribution's flexibility to align with the target's tail behavior, and b) enhancing the flow's training process robustness to adequately capture the tail dynamics of the target distribution.

### 4.1 Tail-Adaptive Base Density for Mixed-Tail Targets

We propose using a mixture of Generalized Gaussian Distributions (GGDs) as the base density of the flow. The Generalized Gaussian represents a family of continuous probability distributions that extend the concept of the Gaussian distribution by incorporating a shape parameter, denoted as $\beta > 0$. Formally, the probability density function (PDF) of a GGD with mean $\mu$, scale parameter $\alpha > 0$, and shape parameter $\beta > 0$ is defined as:

$$f(x; \mu, \alpha, \beta) = \frac{\beta}{2\alpha\Gamma(1/\beta)} \exp\left( - \left( \frac{|x - \mu|}{\alpha} \right)^{\beta} \right)$$

where $\Gamma(\cdot)$ is the gamma function. This distribution encompasses a wide range of density shapes including but not limited to the Gaussian distribution ($\beta = 2$) and Laplacian distribution ($\beta = 1$). Higher values of $\beta$ give rise to light-tailed distributions, whereas lower $\beta$ values lead to heavy-tailed distributions. Figure 2 shows the flexibility of the tail behavior of GGD with different shape parameters (for a more thorough discussion on GGDs, see (Dytso et al., 2018)).

A trainable mixture model as the base distribution is defined as:

$$p_u(u; \pi_1, \ldots, \pi_M, \psi_1, \ldots, \psi_M) = \sum_{m=1}^{M} \pi_m \ p(u; \psi_m)$$

where $m$ is the component index, $\pi_m \in [0, 1]$ is the weight of the $m$th component such that $\sum_m \pi_m = 1$, and $\psi_m$ are the parameters of the $m$th GGD component parameterized by $(\mu, \sigma, \beta)$.

Our choice for using a mixture for better modelling mixed tail behavior is substantiated by previous work that successfully used mixtures to model tailed distributions (Feldmann & Whitt, 1998; Okada et al., 2020; Venturini et al., 2008). Hagemann & Neumayer (2021) show that normalizing flow training can be stabilized in cases (e.g. disjoint support of base and target) by using a mixture for $p_u(\boldsymbol{u})$. We posit that training a flow model on mixed-tail targets will similarly introduce optimization difficulties that may only be exacerbated by having a flexible base density with wide-ranging tail behavior.

**Satisfying the Tail Condition of the Base Density** Jaini et al. (2020) show that for Lipschitz triangular flows to be able to capture the tail of the target distribution, their base density should be *at least* as heavy tailed. When the tails of the base density $p_u$ are fixed, and the determinant of the Jacobian $\left| \det J_{T_\phi^{-1}}(\mathbf{x}) \right|$ is bounded due to the Lipschitz property, it follows that $p_\phi$ cannot exhibit heavier tails than $p_u$, as the transformation $T_\phi$ cannot increase the rate of decay of $p_u$ at its tails. This implies that for $\boldsymbol{u} \sim p_u(\boldsymbol{u})$ with known tail behavior, and $\mathbf{x} \sim p_\phi(\mathbf{x})$, the tail properties of $p_\phi$ are essentially inherited from $p_u$ and limited by the expressiveness of $T_\phi$, ultimately constraining the model's capacity to represent target densities $p_\phi$ with tails heavier than those of the base density $p_u$.

To comply with this requirement, our choice of a flexible GGD mixture base is motivated by the the ability of GGD to control its tail behavior through its shape parameter. We argue that the mixture base density requires only one component to be at least as heavy-tailed as the target density for the flow model to adhere to the requirement of Jaini et al. (2020), while the flexible nature of the mixture will let the flow capture complex mixed-tail behavior of the target density.

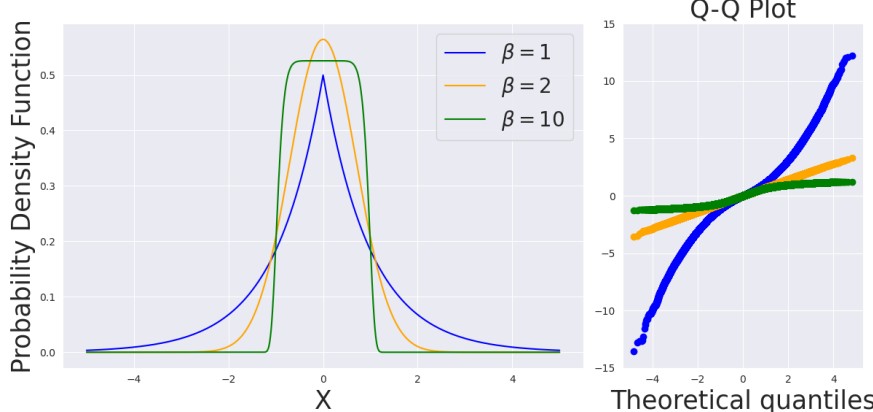

Figure 2: Probability density functions (left) and quantile functions (right) of generalized Gaussian distributions with shape parameter $\beta = 1.0, 2.0$, and $10.0$. $\beta = 2.0$ yields a Gaussian. As $\beta$ drops below $2.0$, the PDF becomes heavy-tailed and the quantile function transitions from linear to concave down and vice versa.

Formally, consider a mixture model $p_u$ where component densities $p(\boldsymbol{u}; \psi_m)$ decay super-exponentially, i.e., $p(\boldsymbol{u}; \psi_m)e^{-c|x|^\alpha}, \alpha \in (0, 2)$, for some $c > 0$. The slowest decaying component ensures the mixture's heavy tail, as $p_u(\boldsymbol{u}; \pi_1, \ldots, \pi_M, \psi_1, \ldots, \psi_M) \sim \pi_m \ p(\boldsymbol{u}; \psi_m)$ as $x \to \infty$.

Considering a mixture of generalized Gaussians, $p(\boldsymbol{u}; \psi_m)$ with $\psi = (\mu_m, \sigma_m, \beta_m)$, a heavy tail is possible if there exists component $j$ which satisfies (i) $\beta_j < \beta_i \ \forall i \neq j$ and (ii) $\beta_j \leq 2$. This indicates the $j$-th component's tail is heaviest and decays super-exponentially. By making $\beta_j$ sufficiently small, we ensure a heavy tail, leading to $p_u(\boldsymbol{u}) \sim \pi_j p(\boldsymbol{u}; \psi_j)$ for large $u$. A single heavy-tailed component is enough to induce heavy tails in the mixture, while the remaining components can vary to adjust the overall density shape. This logic extends to creating arbitrarily light tails. It should be noted that this does not preclude the flow from capturing complex and heterogeneous tail behaviors as would be the case for non-mixture bases. The controllable tail of GGD lets the flow adapt shorter tails as well. The key advantage of using a mixture of GGDs as the base density is that it provides more flexibility to match multimodal marginals with complex tail behaviors, compared to using a single parametric family. Our approach does not make any assumptions about isotropy or impose identical marginal tail decay rates and each mixture marginal can have distinct tail properties.

## 4.2 Training Flows On Mixed-Tail Targets

In the previous section we proposed using a mixture of generalized Gaussians base density for modelling heavy and mixed-tail targets. We also argued that the flexibility to model the tail behavior should come from the flow itself and not be put entirely upon the base density. In this section, based on the motivation provided in the Section 3.2, we propose our method for stabilizing the training process of the flow models by making it tail adaptive.

### 4.2.1 Our Proposed Robust Monte Carlo Maximum Likelihood-Based Training

To mitigate the problem described in previous section, we propose using robust estimation methods during the maximum likelihood estimation. Robust gradient estimators have been employed similarly in the literature (e.g. Hsu & Sabato (2016)). However, to the best of our knowledge they have not been applied to the problem of stabilizing MLE in presence of mixed-tail behavior. Specifically, we propose employing the *Geometric Median (GM)*:

$$\bar{\mathbf{g}} = \arg\min_{\mathbf{g}} \sum_{b=1}^{B} \|\mathbf{g} - \nabla_\theta L(\theta; \mathbf{x}_b)\|_2, \quad \mathbf{g} \in \mathbb{R}^d \tag{2}$$

during the Monte Carlo estimation of the likelihood while training the flow to mitigate the problem of high variance gradients when the target is heavy or mixed-tailed.

To motivate this, we pay attention to mean estimation in presence of heavy-tailed data. Let $X_1, X_2, \ldots, X_n$ be i.i.d. random variables with a heavy-tailed distribution. Let $\bar{X}_n = \frac{1}{n} \sum_{i=1}^{n} X_i$ be the sample mean and $X_{med}$ be the sample geometric median. Then $\mathrm{Var}(\bar{X}_n) \gg \mathrm{Var}(X_{med})$. For heavy-tailed distributions, the variance can be arbitrarily large. In fact, by definition of heavy tails, the tails of the distribution decay slower than an exponential distribution. Therefore, for any finite $V$, there exists some $x_0$ such that:

$$\int_{|x| > x_0} f(x)dx > \frac{V}{n}$$

Where $f(x)$ is the probability density function. This means there is non-negligible probability mass in the heavy tails that can lead to extremely large values of $X_i$. These extreme values disproportionately increase the variance of the sample mean $\bar{X}_n$ (more in depth discussion can be found in literature, e.g. (Sun et al., 2015)). On the other hand, the geometric median only depends on the relative ordering of the $X_i$ values, not their magnitudes. Therefore, it has robust variance in the presence of heavy tails.

Next we posit that using the geometric median estimator for gradients instead of the sample mean when training normalizing flows on mixed-tailed data reduces the variance and leads to more stable optimization. From the training objective, the gradient is:

$$\nabla_\theta \mathcal{L}(\theta) = \mathbb{E}_{p_\mathbf{z}}[\nabla_\theta \log p_\mathbf{x}(f_\theta(\mathbf{z}))]$$

This expectation is typically approximated with Monte Carlo sampling:

$$\nabla_\theta \mathcal{L}(\theta) \approx \frac{1}{n} \sum_{i=1}^{n} \nabla_\theta \log p_\mathbf{x}(f_\theta(\mathbf{z}_i))$$

We already established that when $p_\mathbf{x}$ is heavy-tailed, the gradients $\nabla_\theta \log p_\mathbf{x}(f_\theta(\mathbf{z}))$ have very high variance. Therefore, using the geometric median instead of the sample mean to aggregate the gradients reduces the variance while still being a consistent estimator. This results in more stable optimization. Robust estimators like the geometric median can mitigate the training instability induced by mixed-tailed target distributions for normalizing flows. We demonstrate this phenomenon in Figure 3, which depicts the instability of the standard maximum likelihood estimation in presence of heavy tails. As can be seen, regardless of whether the model is mis-specified or not or the tail behavior of the base and target densities, the loglikelihood estimate is impacted by the heavy-tailed minibatches, resulting in a biased estimation of expectation. This bias is more prominent when the base density is heavy-tailed, which explains why the proposed method of Jaini et al. (2020) yields mixed results in practice. We employ the Weiszfeld method (Eftelioglu, 2017) to estimate the geometric median.

## 5 EXPERIMENTS

We experiment using a range of simulated and tabular data. We train Real NVP (RNVP (Dinh et al., 2017) and Masked Autoregressive (MAF (Papamakarios et al., 2017)) flows with GGD mixture base density (100 components, learnable parameters) and robust MLE. The choice of RNVP is motivated by it being a Lipschitz-continuous function, which results in the density function having the same tail properties as its base, making the dynamics of the model simpler and easier to interpret. The shift and scale operations of RNVPs are modelled by an feed forward network with one hidden layer the width of 1024. The MAF is potentially more expressive than RNVP, letting us study the performance of our method for different flow models. Each flow model is 12 steps deep, with the MAFs having 12 autoregressive layers which use a stacked transformation with 1024 hidden units and 2 blocks to model the conditional dependencies of the variables. Minibatch size is 1024 and we perform Adam optimization ($lr = 10^{-5}$) All trainings are over 1000 iterations. We perform experiments in this section 5 times - unless explicitly mentioned - and report the mean and standard error.

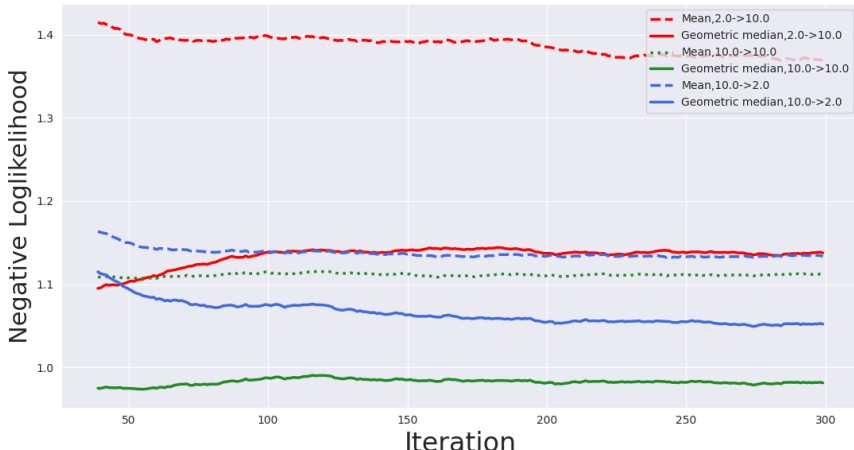

Figure 3: The impact of heavy-tailed target and base on the maximum likelihood estimation. We plot the NLL of a Real NVP flow trained with Student's t base density on a Student's t target using mean (dashed lines) and geometric median (solid lines), with and without tail mis-specification of the base with respect to the target. The legend denotes (degrees of freedom of the base-> degrees of freedom of the target).

## 5.1   Data Quality metrics

To measure the performance of our proposed method we perform experiments on synthetic and real data and report the negative log likelihood. Also following Laszkiewicz et al. (2022) we report the tail-specific performance metrics of Tail Value at Risk (tVaR) and Area Under Log-Log Plot (AULLP). The details of the metrics used can be found in the Appendix C.

**Average Negative Log Likelihood**   In the context of normalizing flows, the negative log-likelihood (NLL) loss is often used as a performance metric to train the model. The NLL loss measures the model's ability to approximate the true data distribution. The negative log-likelihood of the model given the dataset $\mathcal{D}$ is defined as:

$$\mathcal{L}(\theta; \mathcal{D}) = -\sum_{i=1}^{N} \log p_\theta(\mathbf{x}^{(i)})$$

**Tail Value at Risk**   Tail Value at Risk (tVaR) is a measure used to estimate expected losses beyond a certain threshold, known as the Value at Risk (VaR). It calculates the average loss above the VaR level based on a given quantile. The tVaR is obtained by taking the conditional expectation of the loss variable, given that the loss exceeds the VaR level. The VaR itself is determined as the minimum value of the loss variable for which the cumulative distribution function exceeds the specified quantile level. We report the tVaR difference between test dataset and synthetic dataset.

**Area Under Log-Log Plot**   The Area Under Log-Log Plot (AULLP) involves integrating the logarithm of the survival function, which represents the probability of a random variable exceeding a given value, in log-log space beyond a high quantile threshold. We report the The AULLP difference between test dataset and synthetic dataset.

### 5.2   2D Dataset

**Data**   We use a synthetic datasets to demonstrate the performance of our proposed method. For our mixed-tail dataset, following Jaini et al. (2020) we choose our target distribution as a bi-variate Neal's funnel:

$$\mathbf{x}_i = \begin{cases} x_{i,1} \sim \mathrm{N}(\gamma, 1), \\ x_{i,2} \sim \mathrm{N}(0, \ \exp\{x_{i,1}/2 + \lambda\}). \end{cases}$$

To better control the tail behavior of the distribution, we add an additive offset on the variance of the second variable via parameter $\lambda \in \mathbb{R}^{\geq 0}$.

Table 1: General performance metrics for the Neal's funnel for the average NLL as well as AULLP difference and tVaR difference for each variable separately.

| Base Density | NLL $\downarrow$ | AULLP$_{v1}\downarrow$ | AULLP$_{v2}\downarrow$ | tVaR$_{v1}\downarrow$ | tVaR$_{v2}\downarrow$ |
|---|---|---|---|---|---|
| Gaussian | $3.93 \pm .22$ | $8.15 \pm .7$ | $18.09 \pm 1.14$ | $2.4e4 \pm 2e2$ | $1.1e5 \pm 7.6e3$ |
| Mx.Gaussian | $3.86 \pm .2$ | $\mathbf{4.58 \pm .19}$ | $11.15 \pm .56$ | $\mathbf{54.5 \pm 2.6}$ | $36.45 \pm 1.72$ |
| Student's t | $3.86 \pm .39$ | $89.9 \pm 10.9$ | $93.6 \pm 11.6$ | $2e30 \pm 4e27$ | $2e31 \pm 9e28$ |
| Mx.Student's t | $3.88 \pm .2$ | $5.73 \pm .53$ | $14.15 \pm .89$ | $8.8e3 \pm 1e1$ | $8.8e5 \pm 9.2e3$ |
| GGD | $3.83 \pm .31$ | $7.09 \pm .45$ | $13.93 \pm 1.19$ | $81.1 \pm 7.61$ | $161.17 \pm 8.32$ |
| **Our Method** | $\mathbf{3.77 \pm .31}$ | $6.1 \pm .35$ | $\mathbf{5.81 \pm .49}$ | $73.72 \pm 3.73$ | $\mathbf{33.0 \pm 2.62}$ |

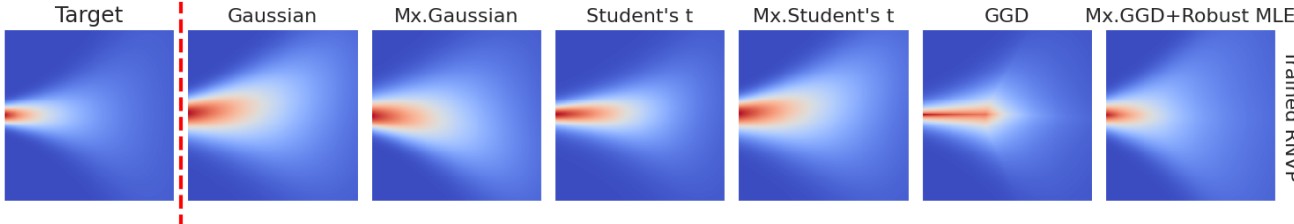

Figure 4: Performance of RNVP flow with different base distributions for estimation of a mixed-tail target. Our modified Neal's funnel with $\lambda = 6.0$ has a heavier-tailed and a lighter-tailed variable.

**Results**   Figure 4 compares the performance of our method for an RNVP flow with baselines with multivariate Gaussian base (as is commonly used in practice), Student's t and generalized Gaussian base densities. Our method performs well for both variables while the other methods' performance is sub-optimal for one or both variables.

Table 1 shows performance of an RNVP with difference base densities. Results reported here show the advantage of our proposed method. We observe a meaningful advantage for our method, especially for the heavier-tailed variable (v2). Of note is the high variance of the flow with Student's t base (following Jaini et al. (2020)) which as discussed in section 3 is due to attenuated tail misspecification in the presence of mixed-tailed targets. The mixture of Gaussian base reports better results for the lighter tailed variable (4.58 vs 6.1 for AULLP, 54.5 vs 73.72 for tVaR)which is to be expected since $v_1$ is a standard Gaussian and has the same normal tail behavior as the flow base. As can be observed in the next section, for targets with more complicated tail behavior, the mixture of Gaussian base loses its advantage.

### 5.3   Ablation Study

**Data**   To measure the impact of each aspect of our method, we perform an ablation study. We use two base distributions, a GGD and a mixture of GGDs. We train an RNVP model on both bases with and without robust MLE. For each base, we train flows on a two-dimensional distribution $P(\mathbb{X}_1, \mathbb{X}_2)$ such that $\mathbb{X}_1$ follows a generalized Gaussian distribution, and $\mathbb{X}_2$ adheres to a Student's t-distribution. $\mathbb{X}$ is parameterized by $\gamma$.

The shape parameter of $\mathbb{X}_1$ is transformed by $\gamma$ via a skew-adjusted Gamma cumulative distribution function, while $\mathbb{X}_2$'s degrees of freedom will be retrieved by $1 + \gamma$. The parameter $\gamma$ is explored within the interval [0, 1000] using a non-exhaustive, selective sampling strategy. Our setup enables us to test out method for a range of distributions with extreme tail behaviors, on one end ($\gamma$=0) having a heavy-tailed Cauchy and a close to uniform short-tailed GGD and at the other end of spectrum ending up with a shorter-tailed Student's t and a Laplace distribution. We train our ablation models for each chosen $\gamma$, and subsequently report the AULLP-diff of two samples from the target and the trained model, each with size 50000.

Figure 5 shows the $\beta$ and $\nu$ parameters of the random variable we use for our ablation study as a function of $\gamma$ (as shown in the two plots on the right). Furthermore, we provide a visual representation of the probability density functions (pdf) for $\mathbb{X}_1$ and $\mathbb{X}_2$ at the extremities of the $\gamma$ spectrum, namely when $\gamma = 0$ and $\gamma = 1000$ (illustrated in the two plots on the left).

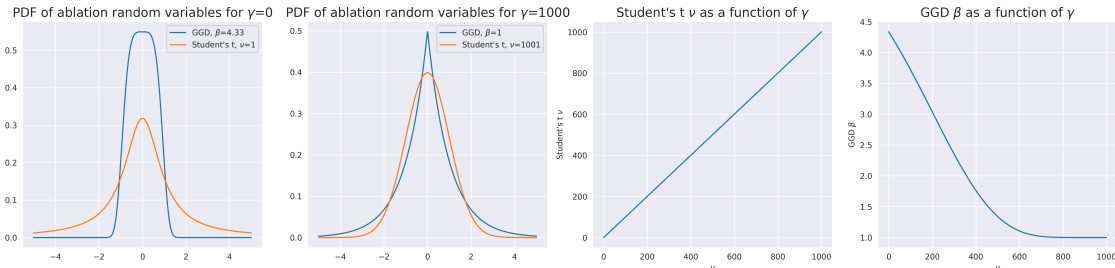

Figure 5: Ablation study framework. The two graphs on the left depict the probability density functions (pdf) of our dual random variables at the extreme points of $\gamma$. The pair of graphs on the right illustrate how the shape parameters — $\nu$ for the Student's t-distribution and $\beta$ for the GGD — vary as functions of $\gamma$.

**Results** The results of our ablation study depicted in Figure 6 show that the flow with mixture of GGD base outperform single GGD in terms of lower AULLP difference across diverse $\gamma$ settings, underscoring the versatility of the mixture in capturing tail behavior of our mixed tail target. Further, robust gradient estimation generally improves model fidelity, although its efficacy varies with the tail behavior of the distributions, which can be attributed to stochasticity of the models - since we do not optimize hyperparameters for different cases individually. These insights reveal that especially for heavier tailed target densities (variable 2) both the GGD mixture base and the robust gradient estimation are needed for the flow to be able to model the tail behavior of the target density accurately.

Table 2: Comparison with (Laszkiewicz et al., 2022). We run the same experiment as Laszkiewicz et al. (2022)'s Table 1 experiment for $d_h = 4$. We repeat our experiment 25 times and report the average results and standard error.

| Method | NLL $\downarrow$ | AULLP-diff$_l\downarrow$ | AULLP-diff$_h\downarrow$ | tVaR-diff$_l\downarrow$ | tVaR-diff$_h\downarrow$ |
|---|---|---|---|---|---|
| TAF | $-8.69$ | 0.42 | 4.05 | 0.89 | 4.36 |
| mTAF | $-8.55$ | **0.25** | 2.6 | **0.57** | 6.74 |
| gTAF | $-8.57$ | 0.5 | 3.38 | 0.98 | 5.55 |
| **Ours** | $\mathbf{-9.03 \pm .08}$ | $0.26 \pm .03$ | $\mathbf{1.53 \pm .21}$ | $0.61 \pm .07$ | $\mathbf{3.8 \pm .49}$ |

### 5.4 Performance Comparison with Pre-Hoc Tail Estimation Methods

In this section we report the performance of our method on a synthetic experiment from Laszkiewicz et al. (2022) to compare the performance of our method to the state of the art in pre-hoc tail estimation-dependent models.

**Data** The target dataset is generated by a 8-dimensional Gaussian copula. The marginals of the copula consist of two Gaussians, followed by a 2-mixture and a 3-mixture of Gaussians. The last four marginals are a mixture of two $t$-distributions with $\nu = 2$. Mixtures have randomized means and variances and equal weights and the correlation matrix is randomized.

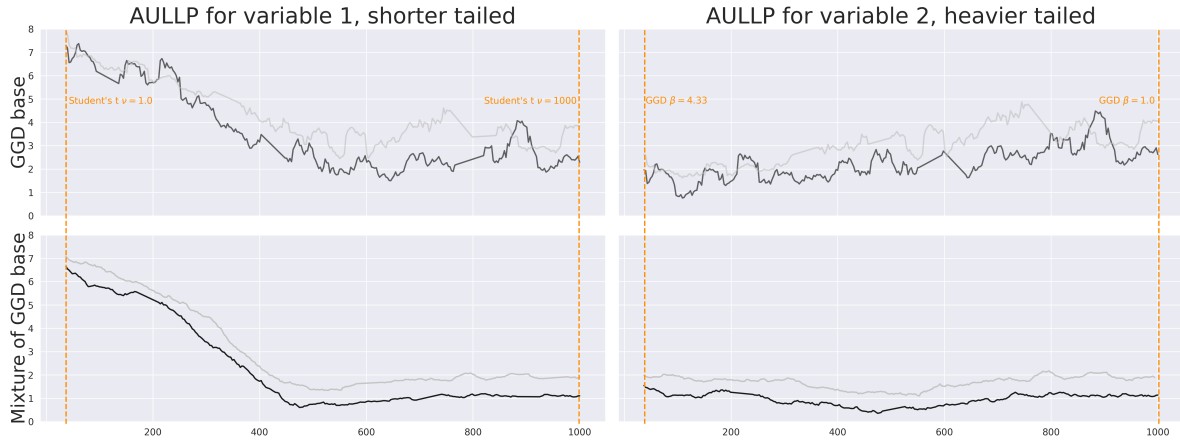

Figure 6: Ablation study results. The AULLP for models trained with two different base distributions are plotted on different rows. Variables $\mathbb{X}_1$ and $\mathbb{X}_2$ are plotted on different columns. The gray lines show the results of models without robust gradient estimation and the black lines show the results with robust gradient estimation. The results show that a)Mixture of GGD provides an improved and more stable performance in capturing the tail behavior of the target, b) robust MLE improves the tail-related performance of the model for both and c) a combination of GGD mixture base density and robust MLE provide the best performance across the board.

More details can be found in Appendix C.2 of Laszkiewicz et al. (2022).

**Results**   Table 2 compares the performance of our method with that of Laszkiewicz et al. (2022). We report the results of training our model on an RNVP - with the same architecture used in previous experiments - with the same setup as per Section 4.1 of Laszkiewicz et al. (2022). Results of Laszkiewicz et al. (2022) are copied from the $d_h = 4$ section of their Table 1. The metrics are reported for heavy tailed and light tailed components separately (denoted by $_h$ and $_l$ subscripts). We observe that our method performs better in case of heavy-tailed components while Laszkiewicz et al. (2022)'s mTAF reports better results for light-tailed components. We attribute our method's slightly lower performance in case of light tails to mis-specification of tails since Laszkiewicz et al. (2022) use Student's t base marginals with the same degree of freedom as the synthetic example ($\nu = 2$) for heavy-tailed target components. This pattern is consistent with results from Section 5.2.

## 5.5   Tabular Datasets

**Data**   To test our method in a more realistic setting, following and expanding on the empirical evaluations done by other methods in this area we train our model on four tabular datasets from the UCI repository, namely *Power*, *Gas*, *Miniboone* and *Hepmass*. We focus on tabular data for the ease with which one can measure and evaluate the tailedness of the training and synthetized data. For other data types such as images and language, due to their inherent high-dimensionality and the rich spatial (for images) and contextual (for language) dependencies it is difficult to define, measure and evaluate the tailedness of the data and the tail-adaptiveness of the model. We follow the same preprocessing steps as Papamakarios et al. (2017) (According to Appendix Section D of their paper). We train RNVP and MAF flows with different base densities as well as our own method. We report negative log likelihood and mean AULLP difference and tVaR difference over all columns.

**Results**   In Table 3, we present a comprehensive evaluation of our proposed approach for an RNVP normalizing flow. The results are contrasted against various base distributions with trainable parameters. We report average tVaR and AULLP over all variables. Our method's superiority is evidenced by the lowest NLL and tVar for all datasets. We report lowest AULLPs for all datasets except MINIBOONE (1.19 vs 1.01). Even

Table 3: Comparison of NLL, AULLP difference and tVaR difference for our proposed method applied to a RNVP in comparison with the same model with different base distributions. R.MLE refers to the robust gradient estimation of Section 4.2.1.
*The tVaR metric tends to grow exponentially when the tail is heavily underestimated

| RNVP + | GAS | | | HEPMASS | | | POWER | | | MINIBOONE | | |
|---|---|---|---|---|---|---|---|---|---|---|---|---|
| | NLL↓ | AULLP↓ | tVaR↓ | NLL↓ | AULLP↓ | tVaR↓ | NLL↓ | AULLP↓ | tVaR↓ | NLL↓ | AULLP↓ | tVaR↓ |
| Gaussian base | $8.86 \pm .56$ | $16.57 \pm 1.32$ | $0.74 \pm .05$ | $10.41 \pm .64$ | $3.38 \pm .22$ | $1.5e2 \pm 9.93$ | $7.8 \pm .47$ | $9.28 \pm .48$ | $2.02 \pm .16$ | $56.29 \pm 4.75$ | $1.34 \pm .10$ | $67.29 \pm 4.07$ |
| Mx.Gaussian base | $8.01 \pm .36$ | $16.51 \pm 1.06$ | $0.72 \pm .04$ | $10.33 \pm .59$ | $3.21 \pm .20$ | $1.21 \pm .08$ | $7.78 \pm .52$ | $9.12 \pm .64$ | $1.79 \pm .10$ | $52.97 \pm 3.58$ | $1.01 \pm .05$ | $65.91 \pm 2.72$ |
| Student's t base | $9.13 \pm 1.53$ | $24.6 \pm 2.74$ | $100.5 \pm 10.14$ | $10.19 \pm 1.38$ | $10.07 \pm 1.19$ | $2.1e9 \pm 3e8^*$ | $10.33 \pm 1.07$ | $21.45 \pm 2.28$ | $2.7e6 \pm 2.4e5$ | $61.63 \pm 6.09$ | $2.05 \pm .28$ | $1.6e3 \pm 2.5e2$ |
| Mx.Student's t base | $7.9 \pm .53$ | $16.6 \pm 1.01$ | $0.72 \pm .06$ | $10.2 \pm .83$ | $3.34 \pm .27$ | $3.38 \pm .29$ | $7.82 \pm .50$ | $9.21 \pm .75$ | $1.73 \pm .17$ | $54.75 \pm 5.02$ | $1.25 \pm .09$ | $66.32 \pm 5.36$ |
| GGD base | $9.02 \pm .82$ | $17.5 \pm 1.96$ | $0.69 \pm .08$ | $10.41 \pm .84$ | $3.38 \pm .39$ | $1.5e2 \pm 11.76$ | $8.45 \pm .89$ | $9.64 \pm .56$ | $2.1 \pm .13$ | $65.39 \pm 6.67$ | $4.37 \pm .27$ | $72.17 \pm 6.45$ |
| **Mx.GGD + R.MLE** | **$7.22 \pm .54$** | **$15.4 \pm 1.67$** | **$0.44 \pm .03$** | **$8.17 \pm .70$** | **$3.18 \pm .30$** | **$1.12 \pm .09$** | **$6.02 \pm .68$** | **$8.64 \pm .76$** | **$1.59 \pm .15$** | **$49.68 \pm 3.74$** | **$1.19 \pm .10$** | **$63.39 \pm 6.77$** |

in this case the difference is relatively small. Notably, the stark tVaR discrepancy, especially in HEPMASS with Student's t base (2.1e9 vs 1.12), underscores the challenge of tail estimation which our method effectively addresses. Our method reports superior results across the board except for tVar in HEPMASS dataset (1.08 vs 1.12), again with a very small margin. This comparison showcases the robustness of our methodology for capturing the tail behavior of the mixed-tail targets. Also of note is the inferior performance of the Student's t and GGD bases. Considering these results alongside results from section 5.2 we conclude that while the Gaussian base maintains a fixed tail behavior and produces average results dependant on the tail behavior of the target distribution and its pathologies, since both Student's t and GGD's tail behavior is parameterized and trainable, the flow either pushes them to underestimate the tail - violating the requirements set by Jaini et al. (2020) - or overestimate the tail, leading to generation of samples with extreme values which will results in subpar performance in tail-specific metrics.

Table 4: Comparison of NLL, AULLP difference and tVaR difference for our proposed method applied to a MAF in comparison with the same model with different base distributions. R.MLE refers to the robust gradient estimation of Section 4.2.1.

| MAF + | GAS | | | HEPMASS | | | POWER | | | MINIBOONE | | |
|---|---|---|---|---|---|---|---|---|---|---|---|---|
| | NLL↓ | AULLP↓ | tVaR↓ | NLL↓ | AULLP↓ | tVaR↓ | NLL↓ | AULLP↓ | tVaR↓ | NLL↓ | AULLP↓ | tVaR↓ |
| Gaussian base | $-3.21 \pm .23$ | $18.07 \pm 1.38$ | $7.8 \pm .70$ | $-54.25 \pm 2.99$ | $13.42 \pm 1.15$ | $17.27 \pm 1.42$ | $2.37 \pm .14$ | $9.45 \pm .72$ | $3.39 \pm .25$ | $-97.12 \pm 6.97$ | $3.86 \pm .26$ | $7.84 \pm .55$ |
| Mx.Gaussian base | $-3.23 \pm .19$ | $18.05 \pm 1.04$ | $7.77 \pm .47$ | $-54.25 \pm 2.39$ | $13.34 \pm .77$ | $17.1 \pm 1.12$ | $2.37 \pm .15$ | $9.38 \pm .61$ | $3.32 \pm .21$ | $-97.12 \pm 5.07$ | $3.87 \pm .19$ | $7.81 \pm .45$ |
| Student's t base | $-1.72 \pm .17$ | $24.35 \pm 2.78$ | $17.03 \pm 1.56$ | $-27.55 \pm 4.33$ | $16.95 \pm 2.84$ | $36.23 \pm 3.57$ | $1.29 \pm .12$ | $15.8 \pm 1.57$ | $14.92 \pm 2.06$ | $-36.12 \pm 3.83$ | $12.9 \pm 2.15$ | $13.06 \pm 1.93$ |
| Mx.Student's t base | $-3.18 \pm .29$ | $18.17 \pm 1.56$ | $7.94 \pm .76$ | $-53.48 \pm 4.36$ | $13.42 \pm 1.15$ | $17.27 \pm 1.22$ | $3.81 \pm .37$ | $9.7 \pm .64$ | $3.11 \pm .32$ | $-95.57 \pm 10.20$ | $3.89 \pm .25$ | $7.11 \pm .70$ |
| GGD base | $-1.36 \pm .16$ | $27.79 \pm 2.04$ | $26.45 \pm 2.04$ | $-51.13 \pm 5.89$ | $13.68 \pm .95$ | $19.08 \pm 1.91$ | **$-1.1 \pm .09$** | $13.47 \pm 1.44$ | $11.57 \pm 1.25$ | $-81.53 \pm 10.37$ | $17.37 \pm 1.69$ | $18.52 \pm 2.40$ |
| **Mx.GGD + R.MLE** | **$-5.95 \pm .58$** | **$15.19 \pm 1.44$** | **$5.36 \pm .63$** | **$-102.4 \pm 7.21$** | **$9.94 \pm 1.08$** | **$8.65 \pm .63$** | $1.57 \pm .14$ | **$6.96 \pm .63$** | **$1.47 \pm .10$** | **$-96.13 \pm 6.77$** | **$3.81 \pm .43$** | $7.93 \pm .95$ |

We also investigate the efficacy of our method on a Masked Autoregressive Flow. Our findings, summarized in Table 4, strongly suggest that integrating the GGD mixture base with robust MLE significantly improves the model's general performance - in terms of Negative Log Likelihood (NLL) - as well as its capabilities to capture tail behavior - as evident from metrics AULLP and tVaR, for which we report the difference.

Our proposed method achieves the lowest NLL for all datasets except POWER and in this exception the margin of difference is quite small. This improvement is most pronounced in the HEPMASS dataset, where the NLL reached a remarkable $-102.4$. Additionally, our approach also yields the smallest AULLP and tVaR values in most datasets, highlighting its capability to more accurately capture tail distributions and overall density estimation.

# 6 DISCUSSION AND FUTURE WORK

In this work we demonstrated the practical utility of using generalized Gaussians for base density coupled with a robust maximum likelihood training algorithm when training normalizing flows to model mixed-tailed data. We proposed a new method of making normalizing flows capable of modelling mixed-tail targets without explicitly binding the tail behavior of the base density to that of the target and instead letting the gradient-based optimization shape the tail of the base. We showcased the general capabilities of our proposed method in estimating mixed-tail targets.

Our method could potentially benefit from being able to set the number of modes in the base mixture adaptively. We varied the number of mixture modes for different tasks and observed an inflection point in performance enhancement with the increase of mixture components, beyond which the model's effectiveness

declines. An extension that uses an infinite mixture model could be a promising future direction. To model more complicated multimodal and asymmetric tail behaviors, a mixture of mixtures structure could be a promising direction for future improvements. Our proposed method also enables, with slight modifications introduced in the Appendix B, selective generation from specific regions of the learned distribution by targeting components of the latent space mixture model, offering finer-grained control over generated samples. Future work should explore this selective generation capability further, including its application in downstream tasks such as fairness-aware machine learning and anomaly detection.

### Acknowledgments

This research has been performed as part of the *Enabling Personalized Intervention* (EPI) project. The EPI project is funded by the Dutch Science Foundation in the Commit2Data program, grant number 628.011.028.

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

## A  Points of Failure in Tail Estimators

Figure 7 shows the accuracy and consistency of the bootstrapped Hill's estimator $\hat{\alpha}_H$ for samples with varying sizes from a Pareto distribution with different tail indices. We report the difference between true and estimated tail indices. We observe that $\hat{\alpha}_H$ overestimates for heavy-tailed samples and underestimates for short-tailed ones. We also observe increased variance for smaller batch sizes – closer to the usual minibatch sizes – from extremely heavy-tailed targets.

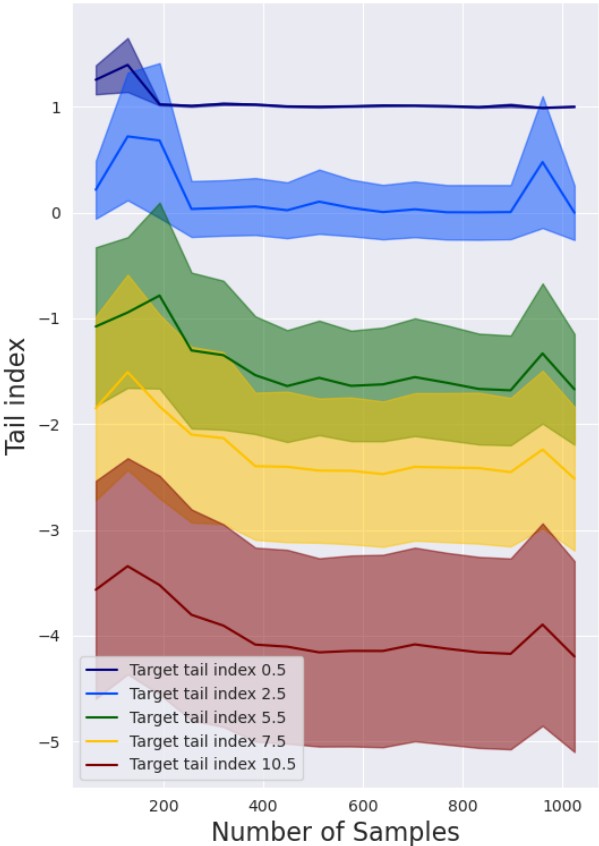

Figure 7: Mis-estimation of tail indices by bootstrapped Hill method for datasets sampled from a Pareto distribution. Tail indices vary from heavy ($\alpha$=0.5) to short ($\alpha$=10.5) and sample sizes from 64 to 2048. Each experiment is conducted 9 times and the mean - solid lines - and standard error - shades - of the difference between the true tail index and the estimated one are reported.

## B  Selective Generation by Exploring the Latent Space

In this section, we introduce an extension on our tail-adaptive normalizing flows method for controlled data generation, enabling precise sampling from specific regions of complex probability distributions. By regularizing the training process to encourage the mixture components to spread out in the latent space, different components specialize in generating different regions of the target distribution, from the tails to the high-density areas around the mean. This allows for targeted sampling from specific components to generate samples from desired regions, offering benefits in applications like fairness-aware machine learning where oversampling minority groups in the tail regions is essential.

We posit that, given our use of a tail-adaptive mixture to model the latent space of our flow model, and considering that each component of the mixture can exhibit a spectrum of tail behaviors, different individual components of the mixture – or subsets thereof – will, upon model convergence, be responsible for generating different parts of the target marginals relative to their tail behavior. In other words, some components will be responsible for generating the tail of the target distribution while alternate components are assigned to represent the regions of high probability density, typically centered around the distribution's mean. To capture extreme tails, we find that it is helpful for the components of the mixture base distribution to be

spread-out. We can encourage this behavior by adding an extra term to the maximum likelihood objective:

$$R(\boldsymbol{\mu}_1, \ldots, \boldsymbol{\mu}_K) = -\frac{1}{K} \sum_{k=1}^{K} ||\boldsymbol{\mu}_g - \boldsymbol{\mu}_k||_2 \tag{3}$$

where $\boldsymbol{\mu}_g$ denotes the geometric median of the component locations ($\boldsymbol{\mu}_k$). Since the means could, due to initialization or the stochastic nature of the optimization, have long tails, we calculate the geometric median of the means, $\mu_g$ and then calculate the sum of the $l_2$ norms of the means and the calculated medians as the total spread of the means. We add the $R$ term to the model's loss function. The model is thereby rewarded during optimization by increasing the distance across the locations.

Our training algorithm including the regularization term will be:

**Input:** $\theta$: flow, $\psi\{\mu, \Sigma, \beta, \pi\}$: base **for** *minibatch* $\mathbf{x}i$ **do**

$\quad R(\boldsymbol{\mu}1, \ldots, \boldsymbol{\mu}K) = -\frac{1}{K} \sum k = 1^K |\mu g - \mu k|2$

$\quad \mathcal{L} = \mathbb{E}\left[p^*(x_i)|p\mathbf{x}(\mathbf{x_i}; \boldsymbol{\theta})\right] + R(\boldsymbol{\mu}1, \ldots, \boldsymbol{\mu}K)$

$\quad \mathbf{g}\mu = \arg\min \mathbf{g} \sum_{k=1}^{n} \|\mathbf{g} - \nabla_\theta L(\theta; \mu k)\|_2$

$\quad \Delta\boldsymbol{\theta} \propto -\nabla\theta\mathcal{L}$

$\quad \Delta\boldsymbol{\psi} \propto -\nabla_\psi\mathcal{L}$

**end**

**Algorithm 1:** Training Algorithm for tail-adaptive normalizing flows with enhanced selective generation

The regularization term in our loss function incentivizes the model to push the means of the mixture modes apart, covering a larger part of the latest space - including the tail area - instead of being centered around certain areas. This by design leads to each sample mapped to the latent space having a small subset of mixture modes as its nearest neighbors. We leverage this aspect of our model for *Selective Generation.*

To generate samples that are on the tail area of the distribution – or near its mean, or in case of complex real world dataset in a specific area of the joint density of a subset of its variables – we can map one or few samples from that area to the latent space, find $k$ nearest mixture modes to these samples – $k$ being a predetermined number or the radius of a hypersphere around the geometric median of our subsample – and then generate our desired number of samples from these modes by sampling from them indiscriminately or by normalizing their weights to form a discrete mixing distribution and forming a separate mixture model.

The advantage of this method is especially evident for areas such as fairness where minority groups are defined in the joint tail section of several features of the dataset. We can upsample any subgroup within our population which has been shown to help mitigate some of the problems with fairness and more generally with performance of ML models. The advantage here is that we are not blindly exploring the latent space. Rather, we are systematically choosing the components in our base distribution with highest probability of being responsible for generating samples similar to our target sample or subsample and sample directly from those components to be pushed forward through the flow model and be transformed into our target.

## B.1 Selective Generation in Practice

Figure 8 illustrates an example of our method's capabilities for selective generation. We train an RNVP using Algorithm 1 with a 20-component GGD mixture base and robust MLE on a target dataset sampled from a two dimensional Student's t with $\nu$=2. We then sample a dataset with the size of 100000 from the Student's t target and evaluate the probabilities of each sample on our target distribution. We bin the probabilities into 15 equal sized bins, with samples belonging to bins with lower probabilities assumed to be further down either tails of the distribution and vice versa. We then evaluate the samples of each bin on every single of the 20 training base mixture components. The component that exhibits the highest average probability value for each bin is assigned as the primary generator for that bin. Since the samples have the highest average probability value when evaluated on that specific component. Therefore, if we sample from that component in the latent space and push the samples through the flow, we will predominantly yield instances that are belong to that specific bin. In Figure 8, for each bin the component with the highest mean probability values

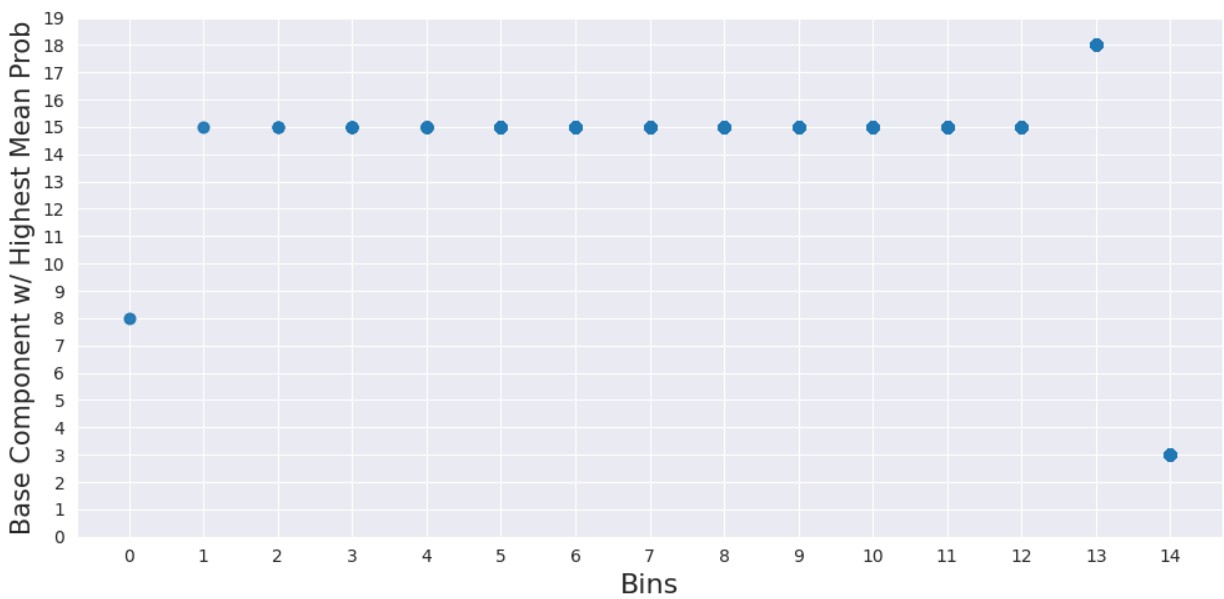

Figure 8: Selective generation from an RNVP model with a 20 component GGD mixture base for a 2d Student's t target.

is marked. We note that component 0 is primarily responsible for generating instances for the bin with the lowest average probabilities (i.e., instances on the tail as evaluated on the target distribution itself), while the majority of the distribution mass is generated by components 3 and 18. Component 15 generates the remaining instances, notably those that are not extreme on the distribution's tail but are situated further from the distribution's mean. It should be noted that our target is a two dimensional target. However, to refrain from over-complicating this analysis we do not discriminate between the dimensions when binning the samples. If each marginal is considered separately, it will offer a more nuanced control over the sample generation process.

In a scenario with a real world target dataset, we will not bin the samples, but rather select a representative sample of the type we want to generate from the target dataset. We then map these representative samples to the latent space. Once we have the mappings in the latent space, we could, for example, choose the top $k$ components with the highest probability values for our selective generation and use a simple weighting scheme (e.g., using the average likelihood values or distance to the component mean), creating a sub-mixture for that particular subset of samples. This sub-mixture will then enable us to upsample our dataset through generation of samples similar to our representative samples.

While not a primary focus of the main paper, the concept of selective generation through targeted sampling within the latent space presents a promising avenue for future research. This approach could be particularly valuable in applications like fairness-aware machine learning, where oversampling minority groups represented in the tail regions of the data distribution is crucial.

## C   Extended Definition of Tail-Related Metrics

**Tail Value at Risk**   The Tail Value at Risk (tVaR) at a quantile level $\alpha$ is an expected loss measure for losses exceeding the Value at Risk (VaR) at the same level, defined as:

$$\text{tVaR}_\alpha(X) = \mathbb{E}[X \mid X > \text{VaR}_\alpha(X)]$$

with $X$ being the loss variable and $\text{VaR}_\alpha(X)$:

$$\mathrm{VaR}_\alpha(X) = \inf\{x \in \mathbb{R} : F(x) > \alpha\}$$

where $F(x)$ is $X$'s cumulative distribution function.

The tVaR-difference metric, the absolute difference between empirical ($\mathrm{tVaR}_\alpha^{emp}$) and model-generated ($\mathrm{tVaR}_\alpha^{model}$) tVaRs:

$$\mathrm{tVaR\text{-}diff}_\alpha = \left| \mathrm{tVaR}_\alpha^{emp} - \mathrm{tVaR}_\alpha^{model} \right|$$

evaluates a generative model's tail capture capability. Smaller tVaR-difference implies better model performance in mimicking the distribution's extreme quantiles.

**Area Under Log-Log Plot**  The Area Under Log-Log Plot (AULLP) involves integrating the logarithm of the survival function, which represents the probability of a random variable exceeding a given value, in log-log space beyond a high quantile threshold. We report the The AULLP difference between test dataset and synthetic dataset.

The Area under the log-log plot (AULLP) measures a model's ability to capture the far distribution tail, emphasizing extreme values more than the Tail Value at Risk (tVaR). It involves integrating the survival function $(S(x) = 1 - F(x))$, which represents the probability of a random variable $X$ exceeding a value $x$, in log-log space beyond a high quantile threshold $x_\alpha$ (e.g., 95th or 99th percentile):

$$\mathrm{AULLP}(X) = \int_{\log x_\alpha}^{\infty} \log S(x)\, d(\log x)$$

To gauge generative models' tail capture accuracy, the AULLP difference (AULLP-diff) between the empirical training data distribution and the model-generated distribution is calculated:

$$\mathrm{AULLP\text{-}diff} = \left| \mathrm{AULLP}^{\mathrm{emp}}(X) - \mathrm{AULLP}^{\mathrm{model}}(X) \right|$$

$\mathrm{AULLP}^{\mathrm{emp}}(X)$ is the empirical distribution's AULLP, and $\mathrm{AULLP}^{\mathrm{model}}(X)$ is for the model-generated distribution.

# D   Experiments

Our training was done on two types of nodes depending on availability, with either one NVIDIA A40 or one A100 GPU.

## D.1   Data preparation

Following the steps from Papamakarios et al. (2017), we prepared the data. The next parts of this section will give a short explanation of the preprocessing done for each dataset.

**POWER**  The POWER dataset (Hebrail & Berard, 2012) includes records of household electricity use over 47 months. Although originally a time series, records were treated as separate, identical samples. The time was changed into an integer representing minutes in the day, and random noise was added. The date and the global reactive power parameter, which often shows a zero value, were removed to prevent unexpected changes in the distribution. Uniform noise was added to each feature. The magnitude of the added noise was large enough to avoid duplication but small enough to keep the data values largely the same. The training data has 1,659,917 examples featuring 6 variables.

**GAS**   The GAS dataset (Fonollosa et al., 2015) contains measurements from 16 chemical sensors exposed to gas mixtures over a duration of 12 hours. Analogous to the POWER dataset, it is a time series that has been handled as if each instance were an i.i.d sample from the marginal distribution. Data was exclusively used from the file *ethylene_CO.txt*, which pertains to an ethylene and carbon monoxide mixture. The removal of highly correlated attributes resulted in an eight-dimensional dataset. The training data has 852,174 examples featuring 8 variables.

**HEPMASS**   The HEPMASS dataset (Baldi et al., 2016) contains particle collision measurements in the field of high-energy physics. Half of the instances denote particle-generating collisions (positive), while the remainder originate from a background source (negative). For this study, positive examples from the "1000" dataset were selected. Five features were excluded due to their high frequency of recurring values, as such repetition can induce spikes in the density and potentially yield misleading outcomes. The training data has 315,123 examples featuring 21 variables.

**MINIBOONE**   The MINIBOONE dataset (Roe et al., 2005), is derived from the MiniBooNE experiment conducted at Fermilab. Like HEPMASS, it contains positive (electron neutrinos) and negative examples (muon neutrinos). For this study, only the positive examples were used. Notable outliers (11 instances) exhibiting a constant value of -1000 across all columns were removed, as well as seven features displaying excessively high counts for specific values, such as 0.0. The training data has 29,556 examples featuring 43 variables.

