# OpenReview forum: "Practical Synthesis of Mixed-Tailed Data with Normalizing Flows"
_TMLR — Accepted by TMLR_

### Review · Reviewer_frCD · 2024-08-04

**Summary Of Contributions:**

The authors address the complex problem of generative modeling on heavy tail data using normalizing flows, with a particular focus on mixed-tail distributions. Recognizing the inherent challenges in selecting or estimating the base distribution for such data, they introduce a novel method for base estimation grounded in generalized Gaussian distributions. This approach aims to more accurately capture the characteristics of mixed-tail data. In their exposition, the authors thoroughly discuss the difficulties and limitations encountered during the training process. To mitigate these issues, they propose the use of a more robust gradient estimator, which enhances the stability and effectiveness of the training phase. The efficacy of their proposed modifications is demonstrated through experiments on both synthetic (toy) models and real-world datasets.

**Audience:**

Yes

**Broader Impact Concerns:**

They are no ethical implications of this work.

**Claims And Evidence:**

Yes

**Requested Changes:**

- Clarify why "the bias of MLE estimator is quite apparent" in Fig. 2
- As mentioned by the authors, (Jaini et al., 2020) mentions that the Lipschitz behavior of triangular maps is a significant limitation. Could we consider non-triangular flows ? Similarly, (Cornish et al., 2020) were also limited by this Lipschitz behavior when working on multimodal data and decided to switch to continuous flows.
- Explain how this method compares to changes made to the transport map itself as in (Hickling et al. 2024)


[Cornish et al., 2020] Cornish, R., Caterini, A., Deligiannidis, G., and Doucet, A. Relaxing bijectivity constraints with continuously indexed normalising flows

[Hickling et al. 2024] Tennessee Hickling, & Dennis Prangle. (2024). Flexible Tails for Normalizing Flows.

**Strengths And Weaknesses:**

**Strengths**
- The paper is well written.
- The intuitive comparison against cited papers is well made.

**Weaknesses**
- As mentioned by the authors, it is not obvious that the mixture base wouldn't introduce significant training challenges in general cases
- Some parameters of the model (namely the weights of the base mixture) lie in a simplex but it doesn't seem like the optimization procedure (simple Adam) takes care of this peculiar geometry.
- Limited questioning on the architecture of the transport map itself.
- How would the proposed geometric median method compare to classic exponential moving average or gradient clipping (which are very popular in the generative modeling community) ? It seems like the training stabilization method hasn't been benchmarked against very natural competitors.
- The experiments show that the method is not very well suited for **mixed** tails data as it underperforms on light tail data compared the heavy tail one.

---

> ### Author Response · Authors · 2024-08-21
> **General rebuttal**
>
> Dear reviewer,
>
> Thank you for your insightful comments. Below are our reactions to several points brought up in your review. We will apply changes to the paper according to your suggestions and update the manuscript once all the reviews are in.
>
> - While standard optimization algorithms like Adam do not inherently account for the simplex constraint, we parameterize the mixture weights using a softmax function.
>
> - In our experiments, we provide results for two different families of flow-based models: RNVPs and non-triangular MAFs. Reporting good results over these two types of affine and autoregressive flows is an indicator that our proposed method is generalizable enough to be used with different flow types. As for the architecture of each model, we perform broad architecture search and hyperparameter tuning as is common practice in similar use-cases.
>
> - EMA, while effective in smoothing gradients and improving convergence, primarily focuses on reducing the impact of noisy gradients, not necessarily addressing the issue of exploding gradients caused by heavy tails. Gradient clipping simply limits the magnitude of gradients, potentially hindering the model’s ability to learn from extreme values present in heavy-tailed data. Our use of the geometric median, however, directly addresses the issue of heavy tails by providing a robust estimator for gradients, mitigating the impact of outliers and stabilizing the training process. This robustness is particularly crucial when dealing with mixed-tail data, where the presence of extreme values can significantly impact the training process.

---

> ### Author Response · Authors · 2024-09-15
> **Further response to reviewer frCD**
>
> Dear reviewer,
>
> Thanks again for your valuable insight. Please find below our response to your requested changes.
>
> 1. This has been addressed by modifying this figure and moving it to Section 4.2.1.
> 2. Jaini et al. (2020) highlight the limitations of Lipschitz behavior in triangular maps and our work leverages this property for its theoretical grounding. Specifically, we utilize RNVP, a triangular flow, as it allows us to directly connect the tail behavior of the base and target densities based on the theoretical framework established by Jaini et al. (2020). However, to demonstrate the generalizability of our approach beyond triangular flows, we also incorporate MAF, a non-triangular and non-affine flow, in our experiments. Our results show that even with MAF, our proposed method effectively captures the target distribution's tail behavior, indicating that the tail-adaptivity stems from our methodological contributions rather than being limited to specific flow architectures. Our results show that our methods work well for discrete time flows and could be extended to other types of non-triangular flows.
> 3. Hickling et al. (2024), which was published in parallel to our submission adds a specific layer to the NF to capture tail behavior. The difference of our method and their proposed method is fundamentally the same as the difference between our method and Laszkiewicz et al’s method, namely relying on some form of tail index estimation to set the parameters of the model (Hickling et al. use bootstrapped Hill). We have discussed the problem with tail index estimation in the paper. Here we also point out that tail index estimation is not possible in a variety of cases, e.g. federated learning and in privacy preserving ML (for example, when we’re using differential privacy and any sort of access to the data incurs a privacy cost).

---

> > ### Comment · Reviewer_frCD · 2024-10-03
> >
> > Thank you for your thoughtful response to my feedback and the feedback from other reviewers. I appreciate the revisions you've made, as they have addressed most of my concerns. However, similar to reviewer kPeN, I remain unconvinced by the experimental results, which do not demonstrate a clear superiority of the proposed method.
> >
> > In addition, I would like to draw your attention to [1], which highlights several limitations of the GM approach. In particular, this paper discusses the computational cost of GM. Could you elaborate on how your method compares in terms of computational cost? Furthermore, could you comment on the other limitations of GM highlighted in [1]? I believe that addressing this recent literature on GM would strengthen the paper and be appreciated by the readership.
> >
> > Overall, I align with reviewer kPeN in that, while I have some reservations, there are no critical limitations that would prevent publication.
> >
> > [1] Fabian Schaipp, Umut Simsekli, & Robert M. Gower (2023). Robust gradient estimation in the presence of heavy-tailed noise. In NeurIPS 2023 Workshop Heavy Tails in Machine Learning.

---

> > > ### Author Response · Authors · 2024-10-05
> > > **Further response to reviewer frCD**
> > >
> > > Dear reviewer,
> > >
> > > Thank you for engaging with us and for your valuable feedback. We appreciate the importance of the points you have brought up. Please find below our response to these points. We hope we have sufficiently addressed your concerns.
> > >
> > > * **On the choice of GM**: Thank you for bringing these points up. We appreciate the reference to the recent work on robust gradient estimation in heavy-tailed noise contexts.
> > > While we acknowledge potential limitations of the Geometric Median approach, including computational costs, our main goal in using GM was to show that some form of gradient stabilization is necessary when dealing with heavy-tailed normalizing flows. This aspect has been largely overlooked in previous heavy-tailed NF research.
> > > Our results show that GM works well for our specific use case. However, we’re not claiming it’s the best solution, and we’re open to the possibility that newer techniques could perform even better, which could be part of future work. As for the practical aspects of employing GM during the training phase, the computational overhead compared to the training itself was negligible in all the experiments we performed.
> > > The key point of our work is highlighting the need for gradient stabilization when training NFs on heavy-tailed or mixed-tail distributions. This insight is new in the field of heavy-tailed NFs.
> > > We agree that expanding our discussion to address GM limitations and compare computational costs would improve the paper. For the camera-ready version we will add a brief discussion acknowledging these limitations. This would provide a more complete picture and suggest directions for future research.
> > >
> > > * **On significance of our results:** About the results, we believe our experimental results *do* demonstrate how our method solves unresolved challenges not addressed by previous work. Our method shows consistent improvements across various scenarios, particularly for heavy-tailed components of mixed-tail distributions. In Section 5.2, we demonstrate superior performance for heavier-tailed variables compared to other base densities, including Student’s t and Gaussian mixtures. Our ablation study in Section 5.3 further supports the efficacy of our approach. In the comparison with state-of-the-art methods (Section 5.4), we outperform existing techniques for heavy-tailed components, while maintaining competitive performance for light-tailed ones (please keep in mind that in our comparison with Laszkiewicz et al. we report results on the same dataset they use which both is an extreme case of the combination of heavy and light tails, and also is sampled from the same distributions as their base density and not, for example, from a Weibull or Log normal, potentially helping their method achieve better results. Also we’d like to point out that the main issue with instability occurs in presence of heavy tails and slight mis-estimation of light tails does not necessarily contribute to degradation of performance during training and inference). This balanced performance across different tail behaviors without making any assumptions about the family of target distributions or relying on tail index estimation techniques is the key strength of our method, especially when dealing with real-world data that often exhibit mixed-tail characteristics. While there may be room for further improvements, we believe our results clearly demonstrate the advantages of our approach in handling complex, mixed-tail distributions in normalizing flows.

---

### Review · Reviewer_kPeN · 2024-08-05

**Summary Of Contributions:**

The paper studies learning of normalizing flows in context of mixed-tail data, introducing a new solution that combines a mixture of generalized Gaussian distributions as a base distribution with a robust training algorithm that uses geometric median of subsample gradients. The method is compared against previous methods proposed for addressing tail behaviour of flows on a few simple data sets with somewhat mixed results. The proposed method is in general best in terms of NLL, but in some experiments is worse in terms of the metric focusing specifically on tails (tVaR and AULLP).

**Audience:**

Yes

**Claims And Evidence:**

No

**Requested Changes:**

The following list includes both concrete requests or recommendations and a few questions.

 1) Clarify the role of Appendix B. If these additional elements were used in the experiments, then you need to explain them already in the main paper. If they were not, you should make it much more clear that Appendix B introduces some additional ideas that could be used but were not. Also, please write the regularization math as proper equations, not in-line within a paragraph.
 2) Clarify the method itself. While you give the mathematical details for both the base distribution and geometric median, it would be near impossible to reproduce the work based on the material provided. I recommend also releasing the code.
 3) Why do you use the word 'synthesis' in the title? For me that term refers to generation of artificial data (e.g. for privacy reasons), but I do not see a particular reason for this paper to emphasise synthesis. You are, after all, proposing a general method for generative modelling applications of normalising flows, right? Your experiments are evaluated in terms of NLL, not in terms of synthetized data.
 4) Section 4.2.1 should cite previous works that considered geometric median at the beginning of the subsection, not at the end. This is important to avoid giving the impression the general idea of using GM for improving robustness of the gradient estimates was a new contribution. I would also strongly recommend mentioning GM explicitly already in the introduction, with proper citations.
 5) The notation is inconsistent. In Section 2 you first claim to use bold letters for random variables and non-bold for observations, yet immediately after that say $\mathbf{x}$ refers to observations and $X$ is used to refer to a random variable.
 6) Section 4.1 talks about platykurtic and leptokurtic distributions, as specific formulation relating to tails. Please introduce this perspective where kurtosis is used for characterising the heaviness of the tails already in Section 2.1.
 7) Check use of \citet{} and \citep{}; there are a few occasions where the wrong command is used.
 8) Figure 1 is illustrative, but I would not *start* the whole Introduction with that. A reader wants to start the reading by reading some actual context, and hence the figure should be placed only on the 2nd page. Also, I would suggest adding one more column here with the proposed mixture as the base distribution. Now the figure only shows the importance of using a suitable base distribution, but with that additional column it would also show that your method works in all cases. Does it? Knowing whether it solves these problem instances is important and showing it in this figure is the easiest way of communicating it for the reader.
 9) I don't see the point of Figure 2. It shows well that the likelihood is the best for the base distribution with the correct tail behaviour, but I do not understand what is the supposed message of the gradient magnitude plot or the visualisation of the progress over iterations. All cases appear quite similar throughout the optimization, and hence I do not see why the plot is as a function of iteration. Maybe think a bit more about what exactly you want to show and then create a more informative figure?
 10) The NLL arrow is in the wrong way in Table 2, right?
 11) Mention in Section 5.1 that the details of the metrics are provided in Appendix C.

**Strengths And Weaknesses:**

The paper is sound and in appears technically correct, but some technical details are missing and hence it is somewhat difficult to evaluate the exact correctness. Even though the problem setup is not new and the previous works have provided more rigorous treatment of the topic, the paper  is likely to be somewhat interesting for practitioners looking for better methods to fit flows in cases of difficult tail behaviour. The solution is not particularly novel and combines two known techniques used previously in the literature (learnable mixtures as base distributions, and geometric median for robust learning for heavy-tailed distributions), but the specific solution has not been presented before. It is adequately evaluated against alternatives and shown to work sufficiently well in practice, but it is worth noting that all experiments are simplified and fairly artificial. Even the supposedly more realistic experiment only considers generic UCI data sets.

Formally, the paper satisfies the evaluation criteria but this is partly because the claims are so weak and indirect ('we highlight and discuss problems' and 'generally favorable performance'). As said above, I do think the paper will be interesting for some readers, but it would benefit from sharper and more scientific writing to contribute better for the general literature. Making the claims more explicit would help in achieving that.

**Strengths:**
 - The paper makes the importance of accounting for tail behaviour very clear, especially in Figure 1
 - The solution is natural and easy to understand, and if open source implementation was provided I presume it would be easy to use in practice

**Weaknesses:**
 - The paper lacks theoretical rigor, especially compared to the most related work. Section 4.1 makes an attempt of theoretical analysis when discussing how the requirements of Jaini et al. (2020) are satisfied for the chosen prior, but this part is not particularly deep and the presentation is too verbal.
 - The empirical performance is not particularly convincing and there is no clear evidence the proposed method is truly useful. It is only compared against the previous method of Laszkiewicz et al. (2022) in one experiment that shows a small numerical improvement, but it is hard for the readers to grasp how significant the improvement is.
 - The actual method description is somewhat lacking. Appendix B mentions a specific regularizer and introduces a new concept of "Selective Generation" that appear to be essential components of the method, yet they are not described in the main paper at all (and there is no reference to Appendix B).
 - No open source implementation. As the paper is in general quite scarce on technical details, I believe we should be able to see the details at least in the code.
 - The writing style is somewhat informal and in general the paper falls short of the quality requirements of top-end venues. There are no clear specific weaknesses and e.g. the most critical citations are included, but the general impression is not that of a highly polished scientific paper and the discussion is not particularly deep.

---

> ### Author Response · Authors · 2024-08-21
>
> Dear reviewer,
> Thank you for your valuable comments. We will apply changes to the paper according to your suggestions and update the manuscript and respond to any remaining issues once once all the reviews are in.

---

> ### Author Response · Authors · 2024-09-15
> **Our response to reviewer kPeN**
>
> Dear reviewer,
>
> Thanks again for your valuable input. Please find below our response to your comments and requested changes.
>
> ## Response to the review
>
>
> **On theoretical rigor**: While prior research like Jaini et al. (2020) has delved deeply into the theoretical aspects of tail behavior in normalizing flows, our work takes a more pragmatic approach, finding that theory does not prescribe what works best in practice. By prioritizing practical application, we identify and provide solutions for the shortcomings of existing theoretical frameworks (namely, in assuming or estimating then fixing the tail index).
>
> **On empirical performance**: The papers we have compared our results against represent the current state-of-the-art in the field. To the best of our knowledge, there are no other works specifically focusing on modeling mixed-tail data using normalizing flows. While the numerical improvements may appear marginal in some instances, it's crucial to recognize their significance within the context of tail modeling. Even small improvements in metrics like AULLP and tVaR translate to substantial differences in the generated data's tail behavior, directly impacting downstream applications sensitive to tail events.
>
> **On selective generation**: We did not use our regularization method for selective generation in the main paper and that is why we moved it to the appendix. Although it’s somewhat orthogonal to the paper’s main claims, we believe it is a promising demonstration of our model’s improved control over the generation process.
>
> **On source code and technical details**: We will publish the full repository for our work, including proper licensing and notebooks for experiments after the publication of the paper. Meanwhile, if the reviewer has any specific questions about the implementation, we’d be happy to share snippets or parts of our implementation anonymously.
>
> **On writing style**: Naturally we will polish the paper further for the camera ready version. Meanwhile, if there are specific instances that the reviewer finds lacking in quality we’d be happy to improve on them.
>
> ## Response to the requested changes
>
> 1. The inline appearance of the algorithm was due to a package conflict and has been fixed. The role of appendix B is addressed accordingly in the appendix.
> 2. The code release has been addressed in the weaknesses section. For the method, we’d be happy to expand and clarify on it if the reviewer finds any specific areas vague and/or unclear.
> 3. This is a valid point. We have changed the title accordingly.
> 4. Changes have been made to address this accordingly.  See the revised draft.
> 5. This issue has been fixed in the revised draft.
> 6. We believe the connection between the beta parameter and tail behavior of the GGD is important and helps better motivate our choice. However, we removed mentions of platykurtic and leptokurtic of the distributions since, although interesting, it is not directly relevant.
> 7. This issue has been fixed in the revised draft.
> 8. Figure 1 was moved to page 2. We can adjust the position for the camera ready version at the Editor’s discretion.
> 9. This has been addressed by modifying this figure and moving it to Section 4.2.1.
> 10. This issue has been fixed in the revised draft.
> 11. This has been addressed in the revised draft.

---

> > ### Comment · Reviewer_kPeN · 2024-09-16
> > **Rebuttal response**
> >
> > Thank you for the response and the revised version, as well as the promise of releasing the code. Overall, the presentation has improved and my main concerns have been addressed, and I think the new title is more descriptive. I still think the theoretical rigor and empirical performance leave a bit to desire, but these are not critical limitations that would as such prevent publication.
> >
> > I am still a bit confused about Appendix B. I now understand it's role a bit better myself, but only because of your response -- the paper is still not clear in communicating it. I recommend you add very clear meta-text at the beginning of Appendix B, explaining what the purpose of the section is and why was it included in the paper. Now you write sentences like *"To capture extreme tails, we find that it is helpful to ..."* and *"We can encourage this behavior by ..."* that suggest these components are part of your solution. Note that I have nothing against including the appendix, but I would like it to be completely clear for the reader why the text is there and how it relates to the rest of the paper. I would also expect to see it mentioned somehow in the paper, e.g. in the Discussion as interesting additional detail and possible future work or perhaps somewhere in Section 3.

---

> > > ### Author Response · Authors · 2024-09-18
> > > **Response to suggestions about Appendix B**
> > >
> > > Dear reviewer,
> > > Thank you for your suggestion. We agree with your suggested improvements and will apply them in the camera ready version - specifically, adding a big-picture/outline section at the beginning of Appendix B giving motivation and a high level overview of selective generation, and mentioning it in the discussion section of the paper.

---

### Review · Reviewer_59rd · 2024-09-05

**Summary Of Contributions:**

This paper proposes a  improved framework for training flows-based models with robust capabilities to capture the tail behavior of mixed-tail data. To this end the paper proposes to use a mixture of Generalized Gaussian Distributions (GGDs) as the base density of the flow. Additionally, the paper proposes using employing the Geometric Median during the Monte Carlo estimation of the likelihood at training time. The proposed method is evaluated on 2D synthetic data and tabular datasets.

**Audience:**

Yes

**Broader Impact Concerns:**

None.

**Claims And Evidence:**

Yes

**Requested Changes:**

The paper can be accepted largely as is.

**Strengths And Weaknesses:**

Strengths:
* The proposed method is novel, simple and interesting.
* The proposed solution is really well motivated. The challenges are clearly laid out and then the proposed method "flows" naturally from the challenges.
* The results on both the 2D synthetic data and tabular datasets are very encouraging.
* The paper includes adequate ablation experiments to highlight the utility of the proposed method.

Weakness:
* The paper uses only two types of flows: Real NVP and Masked Autoregressive flows. It would be interesting to add additional evaluations using nonlinear normalizing flows, e.g., (Ziegler & Rush, 2019). These experiments would further highlight the applicability of the proposed method across flows.

* Lack of results on real-world datasets: The paper only includes results on the tubular datasets. While the results are encouraging it is unclear how mix-tailed these datasets are. It would be helpful to include experiments with additional real-world datasets with mix-tailed data.

---

> ### Author Response · Authors · 2024-09-15
> **Our response to reviewer 59rd**
>
> Dear reviewer,
>
> Thank you for your valuable feedback. Please find below our response to the points brought up in your review:
>
> **On use of other types of normalizing flows:** Jaini et al. (2020) highlight the limitations of Lipschitz behavior in triangular maps and our work leverages this property for its theoretical grounding. Specifically, we utilize RNVP, a triangular flow, as it allows us to directly connect the tail behavior of the base and target densities based on the theoretical framework established by Jaini et al. (2020). However, to demonstrate the generalizability of our approach beyond triangular flows, we also incorporate MAF, a non-triangular and non-affine flow, in our experiments. Our results show that even with MAF, our proposed method effectively captures the target distribution's tail behavior, indicating that the tail-adaptivity stems from our methodological contributions rather than being limited to specific flow architectures. Our results show that our methods works well for discrete time flows and could be extended to other types of non-triangular flows. While incorporating normalizing flows within a VAE decoder, as in Cornish et al. (2020), presents an interesting avenue, integrating VAEs into our framework would convolute the analysis and make it challenging to discern the specific contributions of our proposed method towards tail-adaptivity.
>
> **On experiment datasets:** Tabular data is the exact real world data we’re aiming to model and synthesize using our method. Their tail behavior is unpredictable and often mixed, and it is much easier to evaluate the success of the model in capturing the tail of the target distribution. We have used a set of diverse tabular datasets which are not synthetic (i.e. have been produced in real world environment) and report improvements on learning their tail behavior, which gives us the confidence to extrapolate on the generalizability of our proposed method on other datasets and other types of flows.

---

### Decision · Action_Editor_8Sd2 · 2024-10-17

**Recommendation:** Accept as is

**Comment:**

The paper is an addition to the normalizing flows literature, and during the rebuttal process, the authors addressed most of the reviewers' concerns. Therefore, both the reviewers and I recommend its acceptance.

**Audience:**

Given that normalizing flows are widely used in the machine learning community, the paper’s contributions are likely to be of interest to the TMLR audience.

**Claims And Evidence:**

The paper introduces a novel generative method based on normalizing flows to address target distributions with mixed tails. The proposed approach stands out due to two innovations compared to existing normalizing flow techniques. First, it introduces a new family of base distributions, modeled as mixtures of generalized Gaussian distributions. These base distribution parameters are optimized alongside the parameters of the normalizing flow itself. Second, the method tackles the challenge of heavy tails in data distributions by employing the geometric median estimator in gradient estimation, rather than the traditional mean, to improve robustness.

The methodology is effectively demonstrated through a comprehensive series of illustrations.